# Prevalence of soil-transmitted helminths infection and associated risk factors among residents of Jigjiga town, Somali region, Eastern Ethiopia

Abdlmenur Alewi Sedo[1], Ahmed Zeynudin[2], Tariku Belay[2], Mekdes Mekonen Belay[3], Ahmed Mohammed Ibrahim[4]*, Mohamed Omar Osman[4], Ramadan Budul Yusuf[4], Abdifatah Abdulahi[5,6]

**1** Department of Medical Laboratory Sciences, Institute of Health Science, Jigjiga University, Jigjiga, Ethiopia, **2** Department of Medical Laboratory Sciences, Institute of Health Science, Jimma University, Jimma, Ethiopia, **3** Department of Public Health, College of Medicine and Health Science, Werabe University, Werabe, Ethiopia, **4** Department of Public Health, Institute of Health Science, Jigjiga University, Jigjiga, Ethiopia, **5** School of Medicine, Institute of Health Science, Jigjiga University, Jigjiga, Ethiopia, **6** CIH LMU, Munich, Germany

* Ahmey114baba@gmail.com, ahmedmohammed@jju.edu.et

## Abstract

### Background

One of the tropical illnesses that is often overlooked is soil-transmitted helminths, or STHs. In tropical and subtropical nations, where poor sanitation and contaminated water sources are common, they mostly impact the most vulnerable populations.

### Objective

The aim of this study was to ascertain the prevalence of STHs and related risk factors among the people living in Jigjiga town, Somali region, Eastern Ethiopia.

### Methods

A community-based cross-sectional study was revealed from June 1 to July 21, 2023. Study participants were selected through a multistage sampling method, where households were randomly chosen from the kebeles. A semi-structured questionnaire and observational checklist were used to collect some of the data. A stool sample was collected from each participant, and a single Kato-Katz was performed to detect STHs. Bivariate and multivariate logistic regression analyses were performed, and statistical significance was declared at a level of $p$-value < 0.05 between the outcome and independent variables.

### Results

There were 507 participants in this study, and 90.9% of them responded. STH prevalence was 11.4% overall (95% CI = 9.0, 14.0). With a prevalent parasite species, *A. lumbricoides*

**Data availability statement:** All relevant data are within the manuscript and its Supporting information files.

**Funding:** The author(s) received no specific funding for this work.

**Competing interests:** The authors have declared that no competing interests exist.

was 9.3%, *T. trichiura* was 2.8%, and hookworms were 0.2%. Of the overall positive cases, 93.1% are due to single parasite infections. Independent predictors of STHs included low wealth status (AOR = 3.10; 95% CI = 1.25, 7.75; *p* = 0.015), infrequent hand washing before meals (AOR = 3.19; 95% CI = 1.55, 6.57; *p* = 0.002), earthen floors (AOR = 2.32; 95% CI = 1.12, 4.79; *p* = 0.023), and no drinking water treatment habit (AOR = 5.07; 95% CI = 1.89, 13.57; *p* = 0.001).

## Conclusion

Jigjiga town had a low prevalence of STHs infections. Infrequent hand washing habits before meals, earthen floors, low wealth status, and no habit of treating drinking water were significant associated factors. Health education on handwashing, regular deworming, improved access to clean water and sanitation facilities to reduce the burden of STH effectively.

## Introduction

The most prevalent nematode parasite worms that affect humans are helminths that are spread through the soil. The three main species of soil-transmitted helminths (STHs) are *Ascaris lumbricoides* (*A. lumbricoides*), hookworms (*Necator americanus* and *Ancylostoma duodenale*), and *Trichuris trichiura* (*T. trichiura*) [1]. Due to poor hygiene Soil-transmitted helminthes can be transmitted faco-orally through the ingestion of infective eggs of the parasites from contaminated food as a result of poor hygiene or direct skin penetration by the L3 larva [2,3]. Egg deposition begins when the eggs hatch in the colon, mature, and reach adulthood. The vast number of eggs that adult worms lay are discharged through human feces into the environment, where they grow and become infective for humans. Eggs and larvae of STHs can live for several months in the soil (*A. lumbricoides* and *T. trichiura*) and larvae several weeks (*hookworms*) [4].

Soil-transmitted helminths infection has a substantial global health impact. These parasites infect approximately 1.5 billion people (24% of the world's population) which accounts for 820, 440, and 460 million of *A. lumbricoides*, *T. trichiura*, and hookworms, respectively [5]. According to the World health organization 2022 weekly report, STHs are responsible for the loss of 1.9 million Disability Adjusted Life Years. This report indicates that 260.6 million Preschool-Age Children, 653.7 million School-Age Children, 108 million adolescent girls, 138.8 million pregnant & lactating women are at risk of STHs infection [6]. Soil-transmitted helminths are endemic in over 75% of the nations in Africa. In Sub-Saharan Africa, there are 800 million people, 220 million of whom have at least one STHs [7]. Ethiopia ranked 13 among SSA countries in terms of the burden of STHs. *A. lumbricoides, T. trichiura*, and hookworms infect one-third, one-quarter, and one-eighth of the population respectively. Thus, Ethiopia has the second-highest ascariasis, third-highest hookworms, and fourth-highest trichuriasis infections in Sub-Saharan Africa [7,8].

STH distribution is influenced by a number of variables, but places with limited access to clean water supplies, favorable climates, and locations distinguished by poverty are where they are most common [1,9]. These parasites have an uneven distribution in the population of a given region and tend to cluster among a few individuals or households within a community. This could be differences in infection exposure caused by behavioral or environmental factors [10].

Studies on STHs infections in Ethiopia revealed that infection prevalence varied according to the communities and regions studied. Intestinal parasites, particularly STHs, can thrive in many different locations due to unsafe and inadequate water accessibility, unsanitary living conditions, and improper waste management. In Ethiopia, helminthic infections are a common cause of morbidity [11].

Even though the Ethiopian Public Health Institute (EPHI) conducted a nationwide mapping survey for STHs and schistosomiasis in almost all regions of the country [12]. The mapping didn't encompass the current study area; therefore, the status of STH infection is still unknown. Furthermore, factors that contribute to the spread of STHs throughout the community are not well carried out in some parts of Ethiopia, including Jigjiga town specifically. Deworming programs have not been held in schools or the community due to a lack of epidemiological studies in the Somali region, particularly Jigjiga town, concerning STH infection. Therefore, the current study aims to determine the prevalence and associated risk factors of STHs among residents of Jigjiga town, Somali region, Eastern Ethiopia.

## Materials and methods

### Study area

The study was conducted in Jigjiga town, the capital of the Somali regional State. The town is located in the eastern part of the country, 628 km from Addis Ababa, the capital of Ethiopia. Jigjiga is located at latitude 9.35 °N and longitude 42.8 °E. The mean annual rainfall of 880 mm and the mean annual temperature is 30.4 °C. The average annual humidity is 57.1%. The town has 17,001 households (HH) and the estimated total population is 257,613. The town has 20 kebeles (the smallest administration units). There are six governmental health institutions (three hospitals and three health centers) that provide services for the town and surrounding population.

### Study design and period

A community-based cross-sectional study was conducted from June 1 to July21 2023.

### Source and study population

All residents of Jigjiga town were the source population. Whereas, all randomly selected family members of the household living in six selected kebeles of Jigjiga town, aged ≥ 6 months and willing to provide samples, were included in the study. While those who were seriously ill and unable to respond to the question during the data collection period, who took anthelminthic drugs within 2 months, and who had diarrhoea during the stool sample collection period were excluded.

### Sample size calculation

The sample size was determined by using the single population proportion by using the prevalence of STHs of 20.9% ($P = 0.209$) from a previous community-based cross-sectional study conducted in Bibugn Woreda, East Gojjam Zone [13], considering a 95% confidence level ($Z_{\alpha/2} = 1.96$) and 5% marginal error ($d = 0.05$).

Therefore, the sample size was calculated by using the following formula

$$n = \frac{(Z_{\alpha/2})^2 \times pq}{d^2}$$

where $n$ = sample size; $Z$ = 95% confidence interval (1.96); $P$ = prevalence of disease (0.209); $d$ = marginal error (0.05); $q = 1 - p$, thus,

$$n = \frac{(1.96)^2 \, 0.209 (0.791)}{(0.05)^2} = 254$$

By considering a 10% non-response rate with the initial estimate (i.e., 254/10 = 25.4), the sample size was 279. Since multistage sampling is used to select study participants, a design effect of 2 was used to obtain the final sample size for this study. Accordingly, the final sample size was 558.

## Sampling technique

A multistage sampling technique was used, and sampling was done in Jigjiga town, kebeles, and households (HH) levels. Jigjiga town has 20 kebeles (k) and 17,001 HH. Out of these, 6 kebeles (k-1, k-5, k-6, k-9, k-12, and k-19) were selected by using a simple random sampling technique, and these randomly selected kebeles had 6227 HH. To calculate the sample size for each of the six chosen kebeles, the list of HH members and HH numbers was gained from the health extension workers. This was done as a sampling frame. Then proportional allocation was employed. The computed sample interval was found to be 11 ($K = N/n$). The overall sample size was initially divided among the kebeles by the size of their household. The flow of the study participant's selection among residents of Jigjiga town (Fig 1). Selection of participant's among residents of Jigjiga town, Somali region, Eastern Ethiopia. A systematic sampling method with an 11-interval was used to choose the HHs. If any members of the HH missed two consecutive visits, we made replacements by excluding that HH and by conducting another lottery method for the remaining 10 HHs. One family member of the HHs was selected that fulfilled inclusion criteria and was selected by lottery methods and then included in the study.

## Data collection method

Face-to-face interviews were implemented by using a semi-structured questionnaire and an observational checklist. Data from participants regarding the socio-demographic and Water, Sanitation, and Hygiene (WASH) related factors were collected. Factors Collected via Interview (Face-to-Face Interview with a Semi-Structured Questionnaire) Include Socio-Demographic Factors, Water, Sanitation, and Hygiene (WASH) Related Factors. On the other hand, factors collected via observation (observation checklist) include stool sample collection process, water, sanitation, and hygiene (WASH) infrastructure, and laboratory observations.

## Stool sample collection and processing

Standardized stool cups with applicator sticks were given to study participants for collecting stool samples. And instructions were given to collect sufficient (approximately 2 g) stool samples and not to mix their stool with urine or soil. Only the morning (7:00 to 8:00 am) stool sample was collected to limit the time of day variation in STHs egg shedding. A single stool sample was collected by a trained laboratory technologist. After being brought to the Iman Higher Medical Specialty Clinic laboratory, all stool samples were kept at room temperature. To remove debris, fresh stool samples were compressed through a mesh screen. A small portion of sieved faeces was transferred to the hole of the template on a microscope slide. After filling the hole using a spatula, the template was removed, and the remaining 41.7 mg sample was covered with a piece of cellophane soaked in the glycerol. By using

another slide, cellophane was pressed to make a smear. The cellophane was immersed in a glycerol malachite green solution for 24 h one day before stool processing. The glycerol gradually removes the hookworm eggs over time, causing a rapid clearing of eggs and rendering them undetectable to the slide reading. To overcome this, the slides were examined for hookworm eggs in one hour of preparation. Later observed microscopically (10×, 40× objectives) to identify the parasite's eggs and quantify the intensity of infection by following the KK SOP procedure. The eggs were counted and multiplied by 24, and the number of eggs

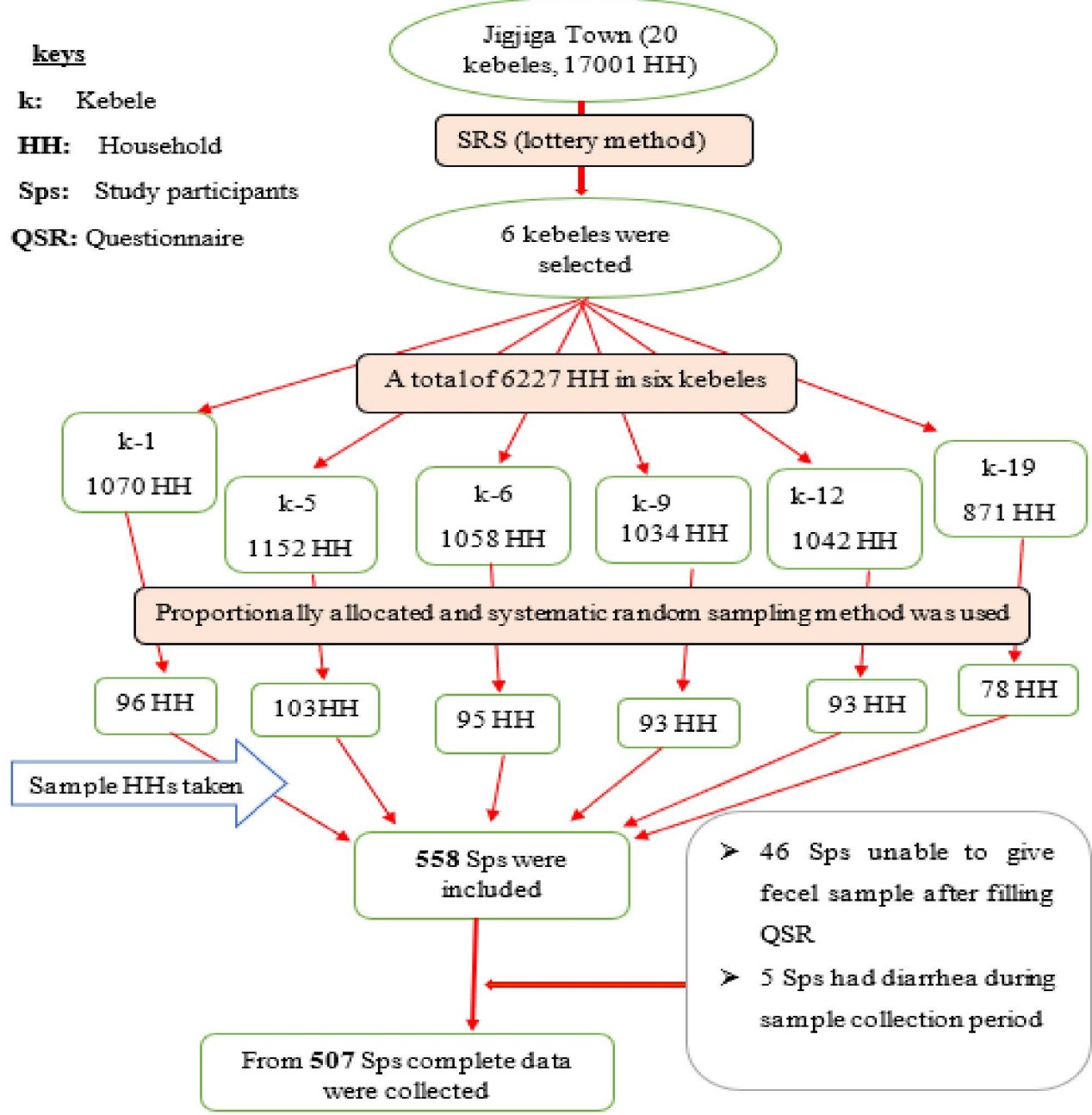

**Fig 1. Flow of the study participant selection process in Jigjiga town, Somali region, Eastern Ethiopia, 2023.**

was expressed as EPG [14]. As recommended by WHO, egg counts were used to classify the intensity of infection as light, moderate, and heavy infection, respectively for *A. lumbricoides* 1–4999epg, 5000–49999epg, and ≥ 50000epg; for hookworm's 1–1999epg, 2000–3999, and ≥ 4000; for *T. trichiura* 1–999epg, 1000–9999epg, and ≥ 10000epg [15].

## Data quality control

To keep the completeness and consistency of the questionnaire, a pre-test was made on 28 individuals (5%) in one Jigjiga town kebele other than the sampled study kebele which was kebele 10. Data collectors composed of laboratory technician and data collectors who had health backgrounds were recruited to collect the data. The training was given to data collectors and laboratory examiners. The functionality of the instruments, the quantity of stool samples, and labelling on collected samples were checked by the supervisor. Kato-Katz smears were examined by two independent senior laboratory technologists. For control purposes, 10% of the examined slides were rechecked by a third senior laboratory technologist.

## Data processing and analysis

Data were cleared, coded, and entered into EpiData (version 4.6.0.6), then exported to and analyzed by SPSS version 26. Frequencies and percentages were calculated to describe the data by tables or figures. Bivariate analysis was performed separately using binary logistic regression to rank the relative importance of exposure variables with outcome variables using odds ratios. The variables that have statistically significant (a *p*-value of < 0.25) associations with the outcome variable were further considered candidates for the multiple logistic regression model to control the effect of confounding variables. Multiple logistic regressions with the backward elimination method were performed. From multiple logistic regressions, exposure variables with a *p*-value <0.05 and a 95% confidence interval were declared as significantly associated factors for STHs. Principal Component analysis (PCA) was used to establish the wealth status of HHs, one of the socioeconomic determinants of STHs infections. By using the PCA, the locally accessible HH asset was considered to calculate HH wealth status. The variables were dummy coded and checked for assumptions like sampling adequacy with Kaiser-Meyer-Olkin, and the results of each analysis were >0.7. The presence of a significant correlation was also confirmed by the correlation matrix, which displays more than two items, and the correlation coefficient was > 0.3. Bartlett's test of sphericity was also examined and found to be significant at *p* < 0.05. After verifying the chi-square assumption, the HH wealth status was calculated and divided into three categories which are low, medium, and high.

**Ethical consideration.** Ethical approval for this study "PREVALENCE OF SOIL-TRANSMITTED HELMINTHS INFECTION AND ASSOCIATED RISK FACTORS AMONG RESIDENTS OF JIGJIGA TOWN, SOMALI REGION, EASTERN ETHIOPIA" was obtained from Institute of Health research and Post Graduate Studies (IHRPGS/821/23). Before data collection, the study was reviewed and approved by the Institute of Health research and Post Graduate Studies (IHRPGS/821/23). Permission was obtained from the Jigjiga Mayor's Office, the Jigjiga town administrative health office, and Iman Higher Medical Specialty Clinic (for stool sample examination). Study participants were included in the study only if the HH head or guardian gives written informed consent for adults and assent from children. The consent request, and assent, available in English and translated into Af-Somali were read entirely to the study participants. To maintain the privacy of participant's interviews were engaged at their homes. Study participants were made aware of the objective of the study. Also, they had the freedom to decline to participate and to leave the study at any moment

without penalty if they so desired. Those study participants who are positive for parasites were linked to the hospital to get treatment without any payment.

## Result

### Socio-demographic characteristics of study participants

Of the 558 study participants, 507 participated in the study, which had a 90.9% response rate. The research participants' ages varied from 1 to 71 years, with a mean age of 21.74 ± 16 SD. Of the study participants, males made up 61.7% (313/507) and 29.2% (148/507) were over the age of 30. About 29.0% (147/507) of participants had completed primary school, while 22.1% (112/507) had no formal education. More than one-third 41.4% (210/507) of the SPs had a low wealth status. The study participants' demographic information was shown in (Table 1).

**Table 1. Socio-demographic characteristics of the study participants in Jigjiga town, Somali region, Eastern Ethiopia, 2023 (*n* = 507).**

| Variable | Category | Frequency | Percent |
|---|---|---|---|
| **Age group in years** | <5 | 81 | 16.0 |
| | 5–9 | 65 | 12.8 |
| | 10–14 | 79 | 15.6 |
| | 15–19 | 46 | 9.10 |
| | 20–24 | 32 | 6.30 |
| | 25–29 | 56 | 11.0 |
| | ≥30 | 148 | 29.2 |
| **Sex** | Male | 313 | 61.7 |
| | Female | 194 | 38.3 |
| **Kebele of residence** | 01 | 90 | 17.8 |
| | 05 | 92 | 18.1 |
| | 06 | 90 | 17.8 |
| | 09 | 85 | 16.8 |
| | 12 | 86 | 17.0 |
| | 19 | 64 | 12.6 |
| **Educational status** | No formal education | 112 | 22.1 |
| | Primary (1–8) | 147 | 29.0 |
| | Secondary (9–12) | 101 | 19.9 |
| | Higher education | 147 | 29.0 |
| **Occupational status** | Gov't employee | 57 | 11.2 |
| | Private employee | 57 | 11.2 |
| | Merchant | 69 | 13.6 |
| | Daily laborer | 52 | 10.3 |
| | Student | 137 | 27.0 |
| | Housewife | 33 | 6.50 |
| | Unemployed | 102 | 20.1 |
| **Wealth status** | Low | 210 | 41.4 |
| | Medium | 132 | 26.0 |
| | High | 165 | 32.5 |
| **Family size** | ≤5 | 299 | 59.0 |
| | >5 | 208 | 41.0 |

## Prevalence of STHs

The study examined 507 faecal samples using the KK smear technique. Of these, the overall prevalence of STHs was 11.4% (95% CI = 9.0, 14.0). The prevalence of *A. lumbricoides* was 9.3% [47/507] 95% CI = 7.0, 12.0], *T. trichiura* was 2.8% [14/507] 95% CI = 1.0; 4.0], and that of hookworms was 0.2% [1/507] 95% CI = 0, 1.0]. From the total of 58 positive cases, a single infection account for 93.1% (54/58). All mixed infections were due to *A. lumbricoides and T. trichiura.* Prevalence of STHs species among residents Jigjiga town (Fig 2). STHs species among residents Jigjiga town, Somali region, Eastern Ethiopia.

The overall prevalence of STHs was the highest among 5–9 age groups, at 20% (13/65), followed by under-5 age groups at 18.5% (15/82). Among study participants, being male had a relatively high chance of being infected with STHs (12.8%; 40/313). Similarly, being single increased the chance of being infected with STHs by 13.1% (37/283). Among participants, residents in kebele 12, 06, and 01 were highly infected with STHs, with a prevalence of 16.3% (14/90), 13.3% (12/90), and 11.1% (10/86), respectively. Study participants who didn't have formal education were highly infected with STHs, with a prevalence of 16.1% (18/112). The prevalence of STH distribution by a sociodemographic factor was presented (Table 2).

## The intensity of STHs

Out of 58 participants who had STH infection, 100% were due to light-intensity infection. No moderate or heavy infection intensity has been detected in the study area. The mean (Egg/gramme) of *A. lumbricoides*, *T. trichiura*, and hookworms was 382 (range: 144–864), 343 (range: 96–792), and 144 (range: 144), respectively. The intensity of STH infections was presented in (Table 3).

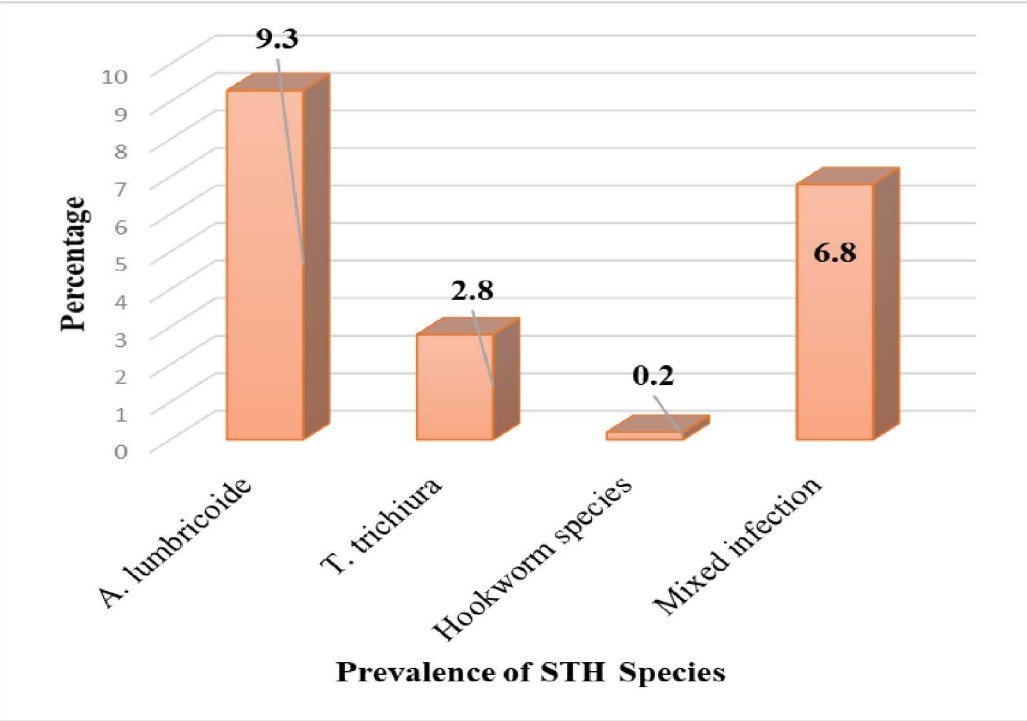

**Fig 2. Prevalence of soil-transmitted Helminth species among residents of Jigjiga town, Somali region, Eastern Ethiopia, 2023.**

**Table 2. Prevalence of STHs distribution by a socio demographic factor among study participants of Jigjiga town, Somali region, Eastern Ethiopia, 2023 (*n* = 507).**

| Variables | Category | The prevalence of STHs | | | |
|---|---|---|---|---|---|
| | | Total positive | A. *lumbricoides* | T. *trichiura* | Hookworm |
| | | *n* (%) | *n* (%) | *n* (%) | *n* (%) |
| **Age groups** | <5 | 15 (18.5) | 11 (13.4) | 5 (6.1) | 0 (%) |
| | 5–9 | 13 (20.0) | 9 (13.8) | 3 (4.6) | 0 (%) |
| | 10–14 | 10 (12.7) | 7 (9.1) | 0 (0) | 1 (1.3) |
| | 15–19 | 6 (13.0) | 5 (11.1) | 0 (0) | 0 (%) |
| | 20–24 | 2 (6.3) | 2 (6.3) | 1 (3.1) | 0 (%) |
| | 25–29 | 3 (5.4) | 4 (7.0) | 1 (1.8) | 0 (%) |
| | ≥30 | 9 (6.3) | 9 (6.1) | 1 (0.7) | 0 (0) |
| **Sex** | Male | 40 (12.8) | 35 (11.2) | 6 (1.9) | 0 (0) |
| | Female | 18 (9.3) | 12 (6.2) | 6 (3.1) | 1 (0.5) |
| **Kebele** | 01 | 10 (11.1) | 0 (0) | 1 (1.0) | 0 (0) |
| | 05 | 10 (10.9) | 1 (1.1) | 0 (0) | 1 (1.1) |
| | 06 | 12 (13.3) | 5 (5.6) | 2 (2.6) | 0 (0) |
| | 09 | 7 (8.5) | 2 (2.4) | 0 (0) | 0 (0) |
| | 12 | 14 (16.3) | 3 (3.5) | 3 (2.8) | 0 (0) |
| | 19 | 5 (7.8) | 1 (1.6) | 0 (0) | 0 (0) |
| **Educational status** | No formal education | 18 (16.1) | 12 (10.7) | 5 (4.5) | 0 (0) |
| | Primary (1–8) | 23 (15.6) | 18 (12.2) | 5 (3.4) | 1 (0.7) |
| | Secondary (9–12) | 7 (6.9) | 7 (6.9) | 1 (1.0) | 0 (0) |
| | Higher education | 10 (6.8) | 10 (6.8) | 1 (0.7) | 0 (0) |
| **Occupational status** | Gov't employee | 4 (7.0) | 4 (7.0) | 0 (0.0) | 0 (0) |
| | Private employee | 4 (7.0) | 4 (7.0) | 0 (0.0) | 0 (0) |
| | Merchant | 4 (5.8) | 4 (5.8) | 0 (0.0) | 0 (0) |
| | Daily laborer | 6 (11.5) | 6 (11.5) | 1 (1.9) | 0 (0) |
| | Student | 18 (13.1) | 14 (10.2) | 4 (2.9) | 1 (0.7) |
| | Housewife | 3 (9.1) | 3 (9.1) | 1 (3.0) | 0 (0) |
| | Unemployed | 19 (18.6) | 12 (11.8) | 6 (5.9) | 0 (0) |

**Table 3. The intensity of STHs infection among residents of Jigjiga town, Somali region, Eastern Ethiopia, 2023 (*n* = 507).**

| Parasites identified | Total infected | The intensity of STHs infection | |
|---|---|---|---|
| | | Mean (egg/gram) | Light intensity |
| A. *lumbricoides* | 47 | 382 | 100% |
| T. *trichiura* | 14 | 343 | 100% |
| Hookworms | 1 | 144 | 100% |

## Binary and multivariable analysis to identify factors associated with the prevalence of STHs infection

For bivariate analysis, variables from socio-demographic, behavioral, and WASH-related factors were included. Among those variables associated with STH infection, eleven variables—sex, kebele, age group, educational status, occupational status, frequency of hand washing before meal, frequency of washing fruits & vegetables before meal, fingernail status, house floor type, wealth status, and drinking water treatment—had a *p*-value <0.25 on bivariate logistic regression analysis, hereafter included in multiple logistic regression analysis for

the backward elimination method. From the total variables included in the multiple logistic regression model, four variables were found to be statistically significant at the level of $p < 0.05$. Hence, the infrequent hand washing habit before meals, house floor type, low wealth status, and drinking water treatment had a significant association with STHs infection.

As a result of this fact, participants who sometimes wash their hands before meals were more than three times more likely to be infected with STHs than those who always wash their hands (AOR = 3.19; 95% CI = 1.55, 6.57; $p = 0.002$). It was also identified that those HH who have an earthen floor were over 2.3 times more likely to be infected with STHs than those who have a cement floor (AOR = 2.32; 95% CI = 1.12, 4.79; $p = 0.023$), and those HH who had low wealth status were three times more likely to be infected with STHs when compared with wealthier (AOR = 3.10; 95% CI = 1.25, 7.75; $p = 0.015$). Moreover, those HHs who didn't have a habit of treating drinking water were five times more likely to be infected with STHs than those who treated water (AOR = 5.07; 95% CI = 1.89, 13.57; $p = 0.001$) (Table 4).

## Discussion

Soil-transmitted helminthiases are caused by infection with the nematode's roundworm, whipworm, and hookworms. Those who live in poverty are particularly susceptible to infection [16]. To advance epidemiological information and target therapies, STH infections must be continuously monitored and evaluated to eliminate morbidity and break transmission of the diseases [17]. Hence, this study attempted to determine the prevalence and associated risk factors among residents of Jigjiga town.

In keeping with the application of mass drug administration, the WHO classifies STHs endemic areas into 3 groups: high transmission (prevalence is greater than 50%), moderate transmission (between 20% and 50%), and low transmission (less than 20%) [18]. According to the results of the current study, 11.4% of subjects tested positive for STHs. This study finding was in line with the researchers conducted in Babile town, Eastern Ethiopia, 13.8% [19], Bamendjou community, Cameroon 11.6% [20]. Barranquilla metropolitan area of Colombia 12% [21], The current prevalence is lower compared with the previously obtained prevalence Thailand's Chachoengsao province 14.3% [22], study in Blue Nile Basins, northwest Ethiopia 30.3% [23], Bibugn woreda, East Gojjam 20.9% [13], Zemika Kebele Bench Maji Zone 70.3% [24], Gilgel Gibe Dam southwest Ethiopia 52.1% [25], Kenya 21.5% [26], Maranhao State, Brazil 23.6% [27], in Ecuador 46.6% [28]. The current prevalence was higher than the results of studies done in the Ijebu-East Ogun State of Nigeria (6.50%) [29], Come, Benin 5.3% [30], Gambia 2.5% [31], 4.8% in Cameroon [17]. In Chinese mainland 4.5% [32], and Zhejiang province in southeast China 1.71% [33].

The above prevalence inconsistency might be a difference in the sample size of the studies, study design, age of study subjects, diagnostic techniques used, and the difference in environmental and climatic factors of the study areas. Furthermore, the differences in awareness of disease transmission between the communities, deworming programs, and variations in worm endemicity in the study sites. For instance, a study in Bibugn Woreda used the two diagnostic techniques kk and formol-ether concentration, and the study area was rural, which may have had a greater impact on the prevalence. Another study was conducted in the Bench Maji Zone, where the average rainfall is 1400 mm³ and the weather condition is Weynadega type, which could be conducive to amplifying the prevalence of parasites. The majority (89%) of study participants in the present study setting had private latrines, which may be the reason for this low prevalence when compared to other studies done in Ethiopia [13,34]. Additionally, the improved health service coverage provided by the local health bureau and health extension workers could be the reason for this low prevalence in the study area.

The most prevalent (9.3%) parasite reported in this study was *A. lumbricoides*, which is lower than the Ethiopian national prevalence of 12.8% [35], Bench Maji Zone, Ethiopia

**Table 4. Bivariate and multivariable analysis to identify factors associated with STHs infection at Jigjiga town, Somali region, Eastern Ethiopia, 2023 (*n* = 507).**

| Variables | Category | Status of STHs | | Bi-variate analysis Multi-variable analysis | | | |
|---|---|---|---|---|---|---|---|
| | | Positive *N* (%) | Negative *N* (%) | COR (95% CI) | *P*-value | AOR (95% CI) | *P*-value |
| Age group | <5 | 15 (18.5) | 66 (81.5) | 3.51 (1.46;8.44) | 0.005* | 3.82 (1.16;12.56) | 0.027 |
| | 5–9 | 13 (20.0) | 52 (80.0) | 3.87 (1.56;9.56) | 0.004* | 1.82 (0.49; 6.71) | 0.371 |
| | 10–14 | 10 (12.7) | 69 (87.3) | 2.24 (0.87;5.77) | 0.095* | 1.42 (.43;4.72) | 0.563 |
| | 15–19 | 6 (13.0) | 40 (87.0) | 2.32 (0.78;6.90) | 0.131* | 3.85 (1.03;14.45) | 0.045 |
| | 20–24 | 2 (6.3) | 30 (93.8) | 1.03 (0.21;5.01) | 0.971 | 1.16 (0.13;10.44) | 0.895 |
| | 25–29 | 3 (5.4) | 53 (94.6) | 0.87 (0.23;3.35) | 0.845 | .722 (.14;3.82) | 0.701 |
| | ≥30 | 9 (6.3) | 139 (93.9) | 1 | | 1 | |
| Sex | Male | 40 (12.8) | 273 (87.2) | 1.43 (0.80;2.58) | 0.231* | 1.83 (.82;4.05) | 0.139 |
| | Female | 18 (9.3) | 176 (90.7) | 1 | | 1 | |
| Kebele of residence | 01 | 10 (11.1) | 80 (88.9) | 1.48 (0.48;4.54) | 0.498 | 0.987 (.23;4.33) | 0.986 |
| | 05 | 10 (10.9) | 82 (89.1) | 1.44 (0.47;4.43) | 0.526 | 1.707 (.41;7.12) | 0.463 |
| | 06 | 12 (13.3) | 78 (86.7) | 1.82 (0.61;5.44) | 0.287 | 0.873 (.21;3.67) | 0.853 |
| | 09 | 7 (8.2) | 78 (91.8) | 1.06 (0.32;3.50) | 0.925 | 0.327 (.059;1.81) | 0.200 |
| | 12 | 14 (16.3) | 72 (83.7) | 2.29 (0.78;6.74) | 0.131* | 1.955 (.46;8.39) | 0.367 |
| | 19 | 5 (7.8) | 59 (92.2) | 1 | | 1 | |
| Educational status | No formal education | 18 (16.1) | 94 (83.9) | 2.62 (1.16;5.94) | 0.021* | 0.436 (.009;21.15) | 0.675 |
| | Primary (1–8) | 23 (15.6) | 124 (84.4) | 2.54 (1.16;5.55) | 0.019* | 1.592 (.23;11.22) | 0.641 |
| | Secondary (9–12) | 7 (6.9) | 124 (84.4) | 1.02 (0.37;2.78) | 0.969 | 2.348 (.50;11.02) | 0.279 |
| | Higher education | 10 (6.8) | 137 (93.2) | 1 | | 1 | |
| Occupational status | Gov't employee | 4 (7.0) | 53 (93.0) | 1 | | 1 | |
| | Private employee | 4 (7.0) | 53 (93.0) | 1.00 (.238;4.21) | 1.000 | 0.331 (.038;2.89) | 0.318 |
| | Merchant | 4 (5.8) | 65 (94.2) | 0.82 (0.19;3.42) | 0.780 | 0.801 (.123;5.24) | 0.817 |
| | Daily laborer | 6 (11.5) | 46 (88.5) | 1.73 (.46;6.50) | 0.418 | 0.283 (.033;2.45) | 0.252 |
| | Student | 18 (13.1) | 119 (86.9) | 2.00 (0.65;6.21) | 0.228* | 0.064 (.005;.780) | 0.031 |
| | Housewife | 3 (9.1) | 30 (90.9) | 1.33 (0.28;6.32) | 0.724 | 1.628 (.222;11.92) | 0.631 |
| | Unemployed | 19 (18.6) | 83 (81.4) | 3.03 (0.98;9.41) | 0.055* | 0.055 (.002;1.49) | 0.085 |
| Hand washing before meal | Always | 22 (7.0) | 293 (93.0) | 1 | | 1 | |
| | Sometimes | 24 (16.4) | 122 (83.6) | 2.62 (1.42;4.85) | 0.002* | 3.19 (1.55;6.57) | 0.002** |
| Washing fruits & vegetables before meal | Always | 19 (7.1) | 249 (92.9) | 1 | | | |
| | Sometimes | 25 (14.5) | 149 (85.6) | 2.20 (1.17;4.13) | 0.014* | 0.602 (.28;1.28) | 0.187 |
| Fingernail status | Trimmed | 33 (9.3) | 323 (90.7) | 1 | | | |
| | Untrimmed | 25 (16.6) | 126 (83.4) | 1.94 (1.11;3.39) | 0.020* | 1.239 (.49;3.084) | 0.645 |
| House floor type | Earthen | 31 (14.9) | 177 (85.1) | 1.76 (1.02;3.06) | 0.043* | 2.32 (1.12;4.79) | 0.023** |
| | Cement | 27 (9.0) | 272 (91.0) | 1 | | 1 | |
| Wealth status | Low | 35 (16.7) | 175 (83.3) | 3.45 (1.62;7.44) | 0.001* | 3.10 (1.25;7.75) | 0.015** |
| | Medium | 14 (10.6) | 118 (89.4) | 2.06 (0.86;4.91) | 0.105* | 1.39 (0.46;4.23) | 0.559 |
| | High | 9 (5.5) | 156 (94.5) | 1 | | 1 | |
| Drinking water treatment | Yes | 10 (5.3) | 179 (94.7) | 1 | | 1 | |
| | No | 48 (15.1) | 270 (84.9) | 3.18 (1.57;6.45) | 0.001* | 5.07 (1.89;13.57) | 0.001** |

COR, crude odd ratio; AOR, adjusted odd ratio; CI, confidence interval.

* denotes variables in the bivariate model that were associated with the dependent variable.

** denotes variables in the final model that were significantly associated with the dependent variable.

16.4% [24], and Ijebu-East Ogun State of Nigeria 16.7% [29]. Recently, the nation has conducted numerous intervention efforts. It might be possible to reduce the prevalence of parasites by implementing health extensions' home-to-home teaching and an open-defecation-free environment. Moreover, the weather and soil types of the area might not be suitable for the development of the egg and larval stages of the parasite. The increased involvement of governmental and non-governmental organizations, including hospitals, clinics, and schools (by giving health education), may also be a contributing factor to this gap. These organizations may have helped to improve the community's health status in the study area over time. On the contrary, this study was higher than the study conducted in Kenya *A. lumbricoides* 0.4% [26], and in Southern Thailand, *A. lumbricoides* was not detected [36]. These discrepancies could be a result of regional and climatic changes as well as the difference in the study individual's hygiene and socio-economic factors, and they may not have good awareness about STHs or in other study areas. Lower may be due to mass drug administration and improving WASH.

The prevalence of *T. trichiura* was 2.8% in the present study, however, this finding is lower than the study found in Barranquilla metropolitan area, Colombia 7.6% [21] and Ecuador 16.5% [28]. The difference may be due to the correlational study design examined in Colombia and the inclusion of specific age groups with follow-up in Ecuador; these may enhance the prevalence and climatic condition of study areas, which may favor the survival of the parasite. The prevalence was higher than the study in Babile town, Eastern Ethiopia, which found a prevalence of 0.2%. This difference may be explained by the fact that the current study focused on all age groups in the community, whereas the previous study on elementary students might not have generalized the community.

In this present study, the prevalence of hookworm was 0.2%. This is in agreement with the study conducted in Babile Town, Eastern Ethiopia, 0.3% [19], but lower than Gilgel Gibe Dam southwest Ethiopia 44.1% [25], Kenya 19.1% [26], Western Uganda 18.5% [37], and Southern Thailand was 10% [36]. This could be because all of the present study participants were urban, and most of them wore shoes when we visited (97%) compared to areas with high hookworm prevalence; this reduces parasite transmission. Rural community participants in studies in Gilgel Gibe Dam in southwest Ethiopia, Kenya, Western Uganda, and Southern Thailand may be more susceptible to hookworm infection since they play in the ground, live, work in agriculture, and travel long distances barefoot.

For parasite transmission, the intensity of STH infections is very important. In the present study, all infected study participants had light infection intensity, and there is no moderate or heavy intensity infection. This is in agreement with the studies conducted in the Bahir Dar Zuria district, Ethiopia [38] and Kandahar, Afghanistan [39] however, moderate and heavy infection intensity was registered in previous research such as [36,40–42] besides low infection intensities, this can be a good indicator that STH infection should be interrupted in our nation. This current result is below the WHO's goal for the elimination of STHs, which is a proportion of moderate-to-heavy STHs infections caused by STHs of 2% [43]. Individuals with low-intensity infections may not typically seek treatment since they don't exhibit any signs or symptoms, which may contribute to parasite spread in the community and environmental contamination [41]. In the current study setting, improved health service coverage provided by the local health bureau may be the cause of this low parasite infection intensity.

Sanitation, good personal and environmental hygiene, and appropriate latrine use all reduce the number of worms in society. Finding the risk factors specific to a certain geographic area and the population aids in developing an effective prevention and control strategy for infection. Associated risk factors for parasite infection can vary from one area to another [43,44]. The findings of this study indicate that practices of personal hygiene and

household-related factors like infrequent hand washing before meals, earthen floor, wealth status, and lack of treating drinking water were significantly associated with STHs infection. This study shows that participants who had a habit of sometimes washing their hands before meals were more likely to be infected with STHs than those who always wash their hands. This present study agrees with studies done in different areas [13,39,41,45,46].

This is due to the possibility of parasites entering people through contaminated drinking water and food if infrequent handwashing habits are practiced before meal.

Those HHs who live on the earthen floor were significantly associated with STH infection among study participants. Study participants who had earthen floors were more likely to have STH infection than those who had cement floors. This present study agrees with the reports found in Kenya, Kogi State, Nigeria, and Puskesmas Moyudan [26,47,48]. Even when it appears to be clean, earthen floors potential to contain STHs eggs.

The HH's wealth status is another factor that affects the prevalence of STH infection. The current study found that individuals with a middle or high income had a lower risk of acquiring an STH infection than those with a lower wealth status. This finding is supported by the 2017 WHO guideline, which indicates that economically disadvantaged populations had higher rates of STH infection and other neglected tropical diseases [16]. In the present study, the prevalence of STHs was high among study participants; those who had low wealth status were three times more likely to be infected with STHs when compared with wealthier people. This study agrees with different studies done in different study areas [37,39,49]. Contrarily, different studies in Thailand show that family monthly income does not have a significant association with STH infection [36,50].

Water, sanitation, and hygiene improvements are considered to be crucial for the long-term, effective control of STHs [51]. Although drinking treated water was not a common practice in the research setting, it does enhance the risk of STHs infection. However, in the current study, those participants who did not have a habit of treating drinking water were more likely infected with STHs compared to those who did. This finding is consistent with studies carried out in two distinct regions of Ethiopia [24,52]. This is contrary to a study conducted in the Bamendjou community, Cameroon, where drinking treated water had no significant association with STH infection [20].

## Limitations

The double KK and other diagnostic methods didn't apply due to financial constraints. Moreover, the city administration refused to provide a map of the study area because of the political instability in the region.

## Conclusion

This study shows the overall prevalence of STHs was low. There was no evidence of any moderate or heavy infection intensity among study participants. Infrequent hand washing before meals, earthen floor, low wealth status of the HH, and lack of treating drinking water were identified as associated risk factors of STHs in the study area. According to the classification of the WHO, the STHs infection seen in the study region might be categorized as a low-risk area. Future studies are important to further investigate.

## Supporting information

**S1 File**. **Data used in analysis for the current study.**
(XLSX)

## Acknowledgments

The author wishes to thank data collectors, study participants, Jimma University, Jigjiga University staffs and Iman Higher Medical Specialty Clinic staff.

## Author contributions

**Conceptualization:** Abdlmenur Alewi Sedo, Ahmed Zeynudin, Tariku Belay, Mekdes Mekonen Belay, Ahmed Mohammed Ibrahim, Mohamed Omar Osman, Ramadan Budul Yusuf, Abdifatah Abdulahi.

**Data curation:** Abdlmenur Alewi Sedo, Ahmed Zeynudin, Tariku Belay, Mekdes Mekonen Belay, Ahmed Mohammed Ibrahim, Mohamed Omar Osman, Ramadan Budul Yusuf, Abdifatah Abdulahi.

**Formal analysis:** Abdlmenur Alewi Sedo, Ahmed Zeynudin, Tariku Belay, Mekdes Mekonen Belay, Ahmed Mohammed Ibrahim, Mohamed Omar Osman, Ramadan Budul Yusuf, Abdifatah Abdulahi.

**Funding acquisition:** Ahmed Zeynudin.

**Investigation:** Abdlmenur Alewi Sedo, Ahmed Zeynudin, Tariku Belay, Mekdes Mekonen Belay, Ahmed Mohammed Ibrahim, Mohamed Omar Osman, Ramadan Budul Yusuf, Abdifatah Abdulahi.

**Methodology:** Abdlmenur Alewi Sedo, Ahmed Zeynudin, Tariku Belay, Mekdes Mekonen Belay, Ahmed Mohammed Ibrahim, Mohamed Omar Osman, Ramadan Budul Yusuf, Abdifatah Abdulahi.

**Project administration:** Abdlmenur Alewi Sedo, Ahmed Zeynudin, Tariku Belay, Mekdes Mekonen Belay, Ahmed Mohammed Ibrahim, Mohamed Omar Osman, Ramadan Budul Yusuf, Abdifatah Abdulahi.

**Software:** Abdlmenur Alewi Sedo, Ahmed Zeynudin, Tariku Belay, Mekdes Mekonen Belay, Ahmed Mohammed Ibrahim, Mohamed Omar Osman, Ramadan Budul Yusuf, Abdifatah Abdulahi.

**Supervision:** Abdlmenur Alewi Sedo, Ahmed Zeynudin, Tariku Belay, Mekdes Mekonen Belay, Ahmed Mohammed Ibrahim, Mohamed Omar Osman, Ramadan Budul Yusuf, Abdifatah Abdulahi.

**Validation:** Abdlmenur Alewi Sedo, Ahmed Zeynudin, Tariku Belay, Mekdes Mekonen Belay, Ahmed Mohammed Ibrahim, Mohamed Omar Osman, Ramadan Budul Yusuf, Abdifatah Abdulahi.

**Visualization:** Abdlmenur Alewi Sedo, Ahmed Zeynudin, Tariku Belay, Mekdes Mekonen Belay, Ahmed Mohammed Ibrahim, Mohamed Omar Osman, Ramadan Budul Yusuf, Abdifatah Abdulahi.

**Writing – original draft:** Abdlmenur Alewi Sedo, Ahmed Zeynudin, Tariku Belay, Mekdes Mekonen Belay, Ahmed Mohammed Ibrahim, Mohamed Omar Osman, Ramadan Budul Yusuf, Abdifatah Abdulahi.

**Writing – review & editing:** Abdlmenur Alewi Sedo, Ahmed Zeynudin, Tariku Belay, Mekdes Mekonen Belay, Ahmed Mohammed Ibrahim, Mohamed Omar Osman, Ramadan Budul Yusuf, Abdifatah Abdulahi.

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
