## [Decision Letter · Decision Letter 0]

18 Nov 2024

PONE-D-24-41131PREVALENCE OF SOIL-TRANSMITTED HELMINTHS INFECTION AND ASSOCIATED RISK FACTORS AMONG RESIDENTS OF JIGJIGA TOWN, SOMALI REGION, EASTERN ETHIOPIAPLOS ONE

Dear Dr. Ibrahim,

Thank you for submitting your manuscript to PLOS ONE. After careful consideration, we feel that it has merit but does not fully meet PLOS ONE’s publication criteria as it currently stands. Therefore, we invite you to submit a revised version of the manuscript that addresses the points raised during the review process.

ACADEMIC EDITOR: The review of the manuscript has been finalized. Although the manuscript is attractive, it needs further revision. Reviewers mentioned that the sampling technique/procedure was missed and should be incorporated, supported with a diagram, unless the manuscript is not suitable for consideration. The abstract, introduction, methodology, results, and discussions should be revised according to the reviewers' comments./>==============================

We look forward to receiving your revised manuscript.

Kind regards,

Alqeer Aliyo Ali, MSc

Academic Editor

PLOS ONE

Reviewers' comments:

Reviewer's Responses to Questions

Comments to the Author

1. Is the manuscript technically sound, and do the data support the conclusions?

Reviewer #1: No

Reviewer #2: Partly

2. Has the statistical analysis been performed appropriately and rigorously? 

Reviewer #1: No

Reviewer #2: No

3. Have the authors made all data underlying the findings in their manuscript fully available?

Reviewer #1: No

Reviewer #2: Yes

4. Is the manuscript presented in an intelligible fashion and written in standard English?

Reviewer #1: Yes

Reviewer #2: Yes

5. Review Comments to the Author

Reviewer #1: I have completed my review of the manuscript and attached my comments and questions for the authors to consider during their revisions. I believe that if the authors address these comments, the manuscript will be suitable for further consideration. I hope to be available to review the revised version of the manuscript.

Reviewer #2: I have completed my review. I found that the manuscript needs extensive revision before further consideration. I recommend the author(s) consider my comments and concerns mentioned below accordingly.

Abstract Method Section:

Add analysis software and statistical significance cutoff value.

Abstract Results:

Write the names of the parasites A. lumbricoides and T. trichiura in italics throughout the document.

Keywords:

Replace "prevalence" and "risk factors" with more suitable words.

Introduction: Write parasite names in scientific format (italicized).

Avoid starting paragraph 3 with an abbreviation.

Methodology:

Inclusion and Exclusion Criteria:

Add this sub-title and separate the study population and criteria.

State whether participants were enrolled based on STH symptoms.

Sampling Information:

Briefly describe how participants were enrolled using multistage sampling.

Include a diagram showing sampling procedures at each stage.

Change the "questionnaire data" sub-title to "data collection method."

List factors collected via interview and observation separately under the data collection method.

Attach both the questionnaire and observational checklist with the revised manuscript.

Results:

Remove the statement "No other intestinal parasites (0%) other than STHs were found in the present study."

Incorporate the Adjusted Odds Ratio and P-value for sex, age group, kebele, educational status, occupational status, fingernail status, and frequency of washing vegetables and fruit in Table 4.

Remove the paragraph stating that certain variables did not confirm a statistically significant association with STH infection.

Discussion:

Explain why studies based on different populations (institutional and rural) were used.

Add limitations of the study at the end of the discussion section.

Include recommendations for future interventions or studies based on findings.

Merge the Ethical Consideration section with the Ethical Approval section to avoid repetition.

6. PLOS authors have the option to publish the peer review history of their article (what does this mean? ). If published, this will include your full peer review and any attached files.

Do you want your identity to be public for this peer review? For information about this choice, including consent withdrawal, please see our Privacy Policy .

Reviewer #1: No

Reviewer #2: No

While revising your submission, please upload your figure files to the Preflight Analysis and Conversion Engine (PACE) digital diagnostic tool, https://pacev2.apexcovantage.com/ . PACE helps ensure that figures meet PLOS requirements. To use PACE, you mus

---

## [Author Response · Author response to Decision Letter 1]

2 Jan 2025

Reviewers Comments to the Authors

MANUSCRIPT: PONE-D-24-41131

TITLE: PREVALENCE OF SOIL-TRANSMITTED HELMINTHS INFECTION AND ASSOCIATED RISK FACTORS AMONG RESIDENTS OF JIGJIGA TOWN, SOMALI REGION, EASTERN ETHIOPIA

Reviewer #1:

General comments

• Reviewer comment: The paper lacks page number.

Author response: Page number is included in the document.

• Parasite names are not written in scientific method (italic) throughout the document.

Author response: Thank you dear reviewer, Names were written italic as you suggested.

• Citation does not align with journal guidelines.

Author response: Thank you dear reviewer, Citations are aligned with journal guidelines.

Abstract

Method: Mention sampling method

Author response: Corrected

Results: “STH prevalence 33 was 11.4% overall (95% CI = 9.0, 14.0)” vague sentence paraphrase it.

Author response: Corrected

Conclusion: You wrote the result again; please add a recommendation based on your finding.

Author response: Corrected

Introduction

• Your citation does not align with the guidelines of the journal. Remove the superscript and write as follows: “[1]”

Author response: Corrected

Methodology

Study area: add health facility information of the Jigjiga Town

Author response: Corrected

Sample technique/Procedures

• The sampling technique is the heart of one study; missing this crucial methodology section indicates that the study wasn’t conducted properly. Also, the readers of the article understand the study clearly. I strongly recommend the author add this section and procedure support with a chart or figure unless this article is rejected.

1. Total kebeles in Jigjiga town

2. Name six selected kebeles of the town by which method.

3. The total eligible population (household) in selected kebeles

4. Source of study participants (household)

5. Proportional allocation (number of participants or household selected from each kebele)

6. Indicate the sampling method applied

7. If systematic, indicate K-value.

8. Mention whether directly selected the participants or their household.

9. As I understood from your study conducted on households of selected kebele, if there were more than one eligible participant in the household, how did you select them?

• Whoever gathers data from the participant, please mention them.

Author response: Based on you recommendation, all these comments in this section were corrected as you suggested by incorporating the necessary information and figure expressing the sampling technique.

• Did the author use the English-language version of the questionnaire for data collection?

Author response: Thank you for your comment. Dear reviewer, Yes since its interviewer administrated questionnaire data collectors know English and collected using English.

• Please indicate it.

• The author mentioned “a pre-test was made on 28 individuals (5%) in one Jigjiga town kebele other than the sampled study study kebele." Name that kebele.

Author response: Based on your comment, we revised and indicated the kebele where we did the pretest.

Results

• Remove this paragraph from the result section and incorporate it into the methodology ology section. “As recommended by WHO, egg counts were used to classify the intensity of infection as light, moderate, and heavy infection, respectively for A. lumbricoides 1-4999epg, 5000-49999epg, and ≥50000epg; for hookworms 1-1999epg, 2000-3999, and ≥4000; for T. trichiura 1-999epg, 1000-9999epg, and ≥10000epg (91).”

Author response: Shifted into the incorporate it into the methodology section.

• The interpretations of AOR are inconsistent, e.g., three times (AOR = 3.19), and 2.3 times (AOR = 2.32). The author should familiarize them and also revise by adding nearly or other appropriate terms.

Author response: Dear Reviewer, we made the correction in response to your feedback.

Discussion

• This study finding is greater than your study result, “Thailand's Chachoengsao province 14.3%.” Check it.

Author response: Thank you for your comment. With all due respect, we rectified it as you suggested and shifted it into the studies that found higher from the current study becouse it higher.

• I recommend the author not repeatedly write results in discussion. AOR results, e.g., (AOR = 2.32; 95% CI = 314 1.12, 4.79; p = 0.023), should be removed from discussion. Data.

Author response: Thank you for your suggestion. We removed it from the discussion.

Declaration

Author contributions

• This sentence was incomplete. “Every author has significantly contributed to the concept, method of study, data collection, analysis, and interpretation of the”

• Similarly, this sentence is also so incomplete. “They also contributed to the manuscript's development, critically reviewed, and decided which publication the paper should be submitted to.”

Author response: Dear reviwer, we appreciate your insight and drawing our attention to completing these sentences and correcting accordingly.

Reviewer #2:

Abstract Method Section:

Add analysis software and statistical significance cutoff value.

Author response: Thank you, we added the analysis software and statistical significance cutoff value.

Abstract Results:

Write the names of the parasites A. lumbricoides and T. trichiura in italics throughout the document.

Author response: Corrected and made italics throughout the document.

Keywords:

Replace "prevalence" and "risk factors" with more suitable words.

Author response: Thank you for your insightful feedback; we corrected it with more suitable words.

Introduction: Write parasite names in scientific format (italicized).

Author response: Corrected and made italics throughout the document.

Avoid starting paragraph 3 with an abbreviation.

Author response: Revised

Methodology:

Inclusion and Exclusion Criteria:

Add this sub-title and separate the study population and criteria.

State whether participants were enrolled based on STH symptoms.

Sampling Information:

Briefly describe how participants were enrolled using multistage sampling. Include a diagram showing sampling procedures at each stage.

Author response: All these comments in this section were corrected as you suggested by incorporating the necessary information based on the comments.

Change the "questionnaire data" sub-title to "data collection method."

Author response: Thank you dear reviewer: Chance is made as you recommended.

List factors collected via interview and observation separately under the data collection method.

Author response: List is incorporated as you mentioned.

Attach both the questionnaire and observational checklist with the revised manuscript.

Author response: The questionnaire and observational checklist with the revised manuscript are attached as you inquired.

Results:

Remove the statement "No other intestinal parasites (0%) other than STHs were found in the present study."

Author response: Removed based on your comment

Incorporate the Adjusted Odds Ratio and P-value for sex, age group, kebele, educational status, occupational status, fingernail status, and frequency of washing vegetables and fruit in Table 4.

Remove the paragraph stating that certain variables did not confirm a statistically significant association with STH infection.

Author response: Dear reviewer, the paragraph you mention is removed from the document.

Discussion:

Explain why studies based on different populations (institutional and rural) were used.

Add limitations of the study at the end of the discussion section.

Author response: Dear reviewer, Agreed and incorporated

Include recommendations for future interventions or studies based on findings.

Author response: Future interventions or studies based on findings to further investigate is included in the recommendation.

Merge the Ethical Consideration section with the Ethical Approval section to avoid repetition.

Author response: Merge was done in the revised manuscript

---

## [Decision Letter · Decision Letter 1]

7 Jan 2025

PREVALENCE OF SOIL-TRANSMITTED HELMINTHS INFECTION AND ASSOCIATED RISK FACTORS AMONG RESIDENTS OF JIGJIGA TOWN, SOMALI REGION, EASTERN ETHIOPIA

PONE-D-24-41131R1

Dear Dr. Ibrahim,

We’re pleased to inform you that your manuscript has been judged scientifically suitable for publication and will be formally accepted for publication once it meets all outstanding technical requirements.

Kind regards,

Alqeer Aliyo Ali, MSc

Academic Editor

PLOS ONE

Additional Editor Comments (optional):

Reviewers' comments:

Reviewer's Responses to Questions

**Comments to the Author**

1. If the authors have adequately addressed your comments raised in a previous round of review and you feel that this manuscript is now acceptable for publication, you may indicate that here to bypass the “Comments to the Author” section, enter your conflict of interest statement in the “Confidential to Editor” section, and submit your "Accept" recommendation.

Reviewer #1: All comments have been addressed

Reviewer #2: All comments have been addressed

2. Is the manuscript technically sound, and do the data support the conclusions?

Reviewer #1: Yes

Reviewer #2: Yes

3. Has the statistical analysis been performed appropriately and rigorously? 

Reviewer #1: Yes

Reviewer #2: Yes

4. Have the authors made all data underlying the findings in their manuscript fully available?

Reviewer #1: Yes

Reviewer #2: Yes

5. Is the manuscript presented in an intelligible fashion and written in standard English?

Reviewer #1: Yes

Reviewer #2: Yes

6. Review Comments to the Author

Reviewer #1: I am satisfied with the authors' responses and corrections. The manuscript is sound and suitable for consideration as it currently stands.

Reviewer #2: The authors revised the manuscript accordingly. Now the paper is appropriate and appreciable for publication in the journal.

7. PLOS authors have the option to publish the peer review history of their article (what does this mean? ). If published, this will include your full peer review and any attached files.

**Do you want your identity to be public for this peer review?** For information about this choice, including consent withdrawal, please see our Privacy Policy .

Reviewer #1: No

Reviewer #2: No

---

## [Editor Report · Acceptance letter]

PONE-D-24-41131R1

PLOS ONE

Dear Dr. Ibrahim,

I'm pleased to inform you that your manuscript has been deemed suitable for publication in PLOS ONE. Congratulations! Your manuscript is now being handed over to our production team.

Kind regards,

on behalf of

Mr. Alqeer Aliyo Ali

Academic Editor

PLOS ONE